



# 1 Application of anisotropy of magnetic susceptibility (AMS)
# 2 fabrics to determine the kinematics of active tectonics:
# 3 Examples from the Betic Cordillera, Spain and the northern
# 4 Apennines, Italy.

David J. Anastasio[1], Frank J. Pazzaglia[1], Josep M. Parés[2], Kenneth P. Kodama[1], Claudio Berti[3],
James A. Fisher[1], Alessandro Montanari[4], Lorraine K. Carnes[5]
[1] Department of Earth and Environmental Sciences, Lehigh University, Bethlehem, PA, 18015-3001, United States
[2] Geochronology, Centro Nacional de Investigación de la Evolución Humana (CENIEH) Burgos, 09002, Spain
[3] Idaho Geological Survey, Moscow, ID, 83844-3014, United States
[4] Osservatorio Geologico di Coldigioco, Apiro MC, 62021, Italy
[5] Arizona State University, Tempe, AZ, 85281, United States
*Correspondence to:* David J. Anastasio (dja2@lehigh.edu)
**Abstract:** The anisotropy of magnetic susceptibility (AMS) technique provides an effective way to
measure fabrics and in the process, interpret the kinematics of actively deforming orogens. We collected rock fabric
data of alluvial fan sediments surrounding the Sierra Nevada massif, Spain, and a broader range of Cenozoic
sediments and rocks across the northern Apennine foreland, Italy, to explore the deformation fabrics that contribute
to the ongoing discussions of orogenic kinematics. Sierra Nevada is a regional massif in the hinterland of the Betic
Cordillera. We recovered nearly identical kinematics regardless of specimen magnetic minerology, structural
position, crustal depth, or time. The principal elongation axes are NE-SW in agreement with mineral lineations,
regional GPS geodesy, and seismicity results. The axes trends are consistent with the convergence history of the
Africa-Eurasia plate boundary. In Italy, we measured AMS fabrics of specimens collected along a NE-SW corridor
spanning the transition from crustal shortening to extension in the northern Apennines. Samples have AMS fabrics
compatible only with shortening in the Apennine wedge and have locked in penetrative contractional fabrics, even
for those samples that were translated into the actively extending domain. In both regions we found that specimens
have a low degree of anisotropy and oblate susceptibility ellipsoids that are consistent with tectonic deformation
superposed on compaction fabrics. Collectively, these studies demonstrate the novel ways that AMS can be
combined with structural, seismic, and GPS geodetic data to resolve orogenic kinematics in space and time.

## 33 1 Introduction

A number of *circum*-Mediterranean orogens are associated with rapid slab rollback, resulting in paired
compressional and extensional domains in the orogenic wedge of the retreating upper plate (Elter, 1975, Carminati
and Doglioni, 2012). Examples include, the Calabria Arc-Tyrrhenian Sea (Beccaluva et al. 1985; Milia et al 2009),
the Hellenic Arc-Aegean Sea (Pichon and Angelier 1979; Papazachos et al. 2000), and the Gibraltar Arc-Alboran
Sea (Lonergan and White 1997; Platt et al. 2006; Fernández-Ibáñez and Soto 2008). Along these tectonic
boundaries, the temporal and spatial relationship between thrust belt contraction, wedge-top basin evolution,



Solid Earth Discussions Open Access EGU

hinterland extension, and orogenic uplift are the subjects of continuing controversy.
Finite and incremental strain data provide deformation history and fabric distribution information for
kinematic studies of folds, faults, and orogens (e.g., Cloos, 1947; Ramsay and Huber, 1984; Fischer et al., 1992).
However, in orogenic forelands where deformation occurs at shallow depths and low temperatures, ductile
penetrative deformation features may be absent and brittle structures may be sparse. Anisotropy of magnetic
susceptibility (AMS) results offers an alternative proxy for grain preferred orientation, and hence rock strain, to
determine the tectonic fabric in these orogens where other deformation markers are not available (Borradaile and
Jackson, 2004; 2010; Borradaile and Henry, 1997; Averbuch et al., 1992; Pares, 2004). In general, comparative
studies from siliciclastic rocks show good agreement between both the relative magnitude and orientation of
penetrative rock strain determined by traditional geometric methods and AMS principal axes, but the AMS axes in
specimens dominated by diamagnetic magnetic mineral abundance, the AMS axes orientations and not necessarily
the magnitude correlates to the rock strain. (e.g., Latta and Anastasio, 2007; Burneister et al., 2009). In this paper,
we show how AMS can extend the temporal reach of GPS geodesy and seismic first motion assessments back in
time in orogenic studies of the Betic Cordillera, Spain and in the northern Apennines, Italy (Fig. 1).

**2 Kinematic Studies For Active Tectonic Research**
Sedimentary rocks acquire a primary depositional fabric, which is bedding-parallel. It is measurable with
the AMS technique and is further enhanced and modified during burial, compaction, and water loss (e.g., Tarling
and Hrouda, 1993; Schwehr et al., 2006). Even unconsolidated rocks record a magnetic fabric that can potentially
provide a kinematic record (Mattei et al., 1997; Porreca and Mattei, 2012). The sensitivity of AMS allows its use is
as a paleogeodetic tool in tectonic studies. Kinematics allow for an assessment of rheology and strain history that are
necessary prerequisites for understanding geodynamics, incrementally balancing cross sections, or in
paleogeographic reconstructions. We sampled both consolidated sedimentary rocks and unconsolidated sediments in
the Betic Cordillera, Spain and northeastern Apennine ranges, Italy for AMS analysis. The Betics field sampling was
designed to test AMS recovery from unburied and unconsolidated sediments around the Sierra Nevada massif. Here,
oriented samples were collected from sites at all structural positions around the Sierra Nevada massif in Plio-
Pleistocene terrestrial, siliciclastic deposits (Table A1). The Apennines field sampling was designed to measure the
rotation of strain across the foreland as rock passes from the actively shortening part of the orogenic wedge near the
trench to the actively extending regime further to the southwest. Here, oriented samples were collected from sites
along a NE-SW oriented corridor inclusive of Cenozoic (Table A1) marine and fluvial siliciclastics, marls, and
carbonate rocks, and unconsolidated Pleistocene fluvial sediments.

3 **The AMS Method**
The AMS ellipsoid is defined by the principal axes ($k_1$-maximum, $k_2$-intermediate, $k_3$-minimum) of a specimen.
It can be represented by a second-rank tensor that characterizes a material's magnetization response to an applied
magnetic field (e.g., Borradaile and Tarling, 1981; Tarling and Hrouda, 1993). The orientation and relative length of
the principal anisotropy axes of a specimen are controlled by the preferred alignment of the anisotropy axes of the



individual magnetic particles in the specimen and the degree of the individual particle's anisotropy. The anisotropy
of individual magnetic grains is controlled by their crystallography and grain shape (Tarling and Hrouda, 1993). For
magnetite grains, the anisotropy is controlled by grain shape, whereas for hematite and phyllosilicate particles the
anisotropy is controlled by crystallography, which, in turn, controls their shape.
Natural processes such as current deposition, lithification, and tectonic deformation all contribute to a
specimen's AMS. In deformed rocks, it was shown that the principal susceptibility axis ($k_1$) orientation is typically
parallel to the strain long axis and orthogonal to the tectonic shortening direction, whereas the shortest axis ($k_3$), is
orthogonal to bedding in orientation (e.g., Kligfield et al., 1982; Hrouda, 1982), regardless of whether the individual
particle anisotropy is controlled by crystallography or shape.
The sedimentary rocks and deposits in this study contain enough phyllosilicate minerals to be excellent
specimens for AMS studies because of the presence of oblate mineral grains which adjust readily to deposition,
lithification, and any subsequent deformation.  As grains reorient in response to depositional or tectonic processes,
the magnetic fabric will continuously adjust (Parés and van der Pluijm, 2002). Deposition from currents in alluvial
fans or rivers like the examples discussed here, will cause preferred grain alignment.  Because the intermediate and
maximum AMS axes of platy grains, such as phyllosilicates, are nearly equal in magnitude they will be randomly
oriented in bedding, with the minimum axes orthogonal to bedding. In mudstones and fine grained sandstones,
where both paramagnetism and ferromagnetism contributions were quantified, paramagnetic mineral grains typically
dominate the AMS signal (Martin-Hernández and Hirt, 2001) because of the shape anisotropy of clay minerals,
although very fine magnetic particles attached to the clay fabric might also contribute (Kodama and Sun, 1992).

**4 Example I: Sierra Nevada Massif, Spain**
**4.1 Geologic Setting of Sierra Nevada Massif**
The Sierra Nevada massif is part of the Betic Cordillera-Rif-Tell orogens that extend along the European-
African plate boundary from the southern Iberian peninsula to northern Africa. These orogens formed by slab
rollback and western migration of the Gibraltar Arc throughout the Neogene (Rosenbaum et al., 2002). Coincident
with the translation of the arc, the upper plate experienced shortening, the growth of doubly-vergent thrust belts,
crustal thickening, and rock uplift (Duggen et al., 2003; Soto et al., 2008;  Platt et al., 2013). In the Betics,
contraction across the plate boundary was initially directed northward (Sanz De Galdeano, 1990; Lonergan, 1993;
Platt et al., 2013). As the contraction continued into the foreland during the late Miocene, it slowed and
progressively rotated to the northwest into its present orientation (Mazzoli and Helman, 1994; Rosenbaum et al.,
2002). Active tectonics in the Betic Cordillera today is dominated by distributed NW-SE convergence of 4-6 mm/yr
(Fernandez-Ibanez et al., 2007; Koulali et al., 2011; Gutshcer et al. 2012; Mancilla et al., 2013) and is
accommodated in part on NW-SE trending normal faults (Martínez- Martínez et al., 2006; Stich et al., 2006;
Fernández-Ibáñez and Soto, 2008; Giaconia et al., 2014; 2015; Fig. 2).
The Sierra Nevada massif is a doubly-plunging, actively uplifting (Azañón et al., 2015) elongate dome,
characterized by medium to low-grade metamorphic rocks stacked in north verging thrust sheets (Martinez-Martinez
and Soto, 2002). Previous interpretations are that the Sierra Nevada dome was uplifted following top to the west



extension and isostatic rebound after thrust belt formation (Martinez-Martinez et al., 2006). Alternatively, as many
culminations exist in orogenic hinterlands, the massif could have been uplifted during contractional or transpressive
strain (e.g., Bernini, 1990; Mitra et al., 1997).
To resolve whether the uplift of the Sierra Nevada dome was the result of extensional exhumation or a
compressional orogenic culmination, we collected rock fabric (AMS) data in Plio-Pleistocene deposits around the
massif to explore the presence of penetrative tectonic fabrics that can contribute additional constraints to the
kinematics of dome emplacement. We focused sampling on unburied alluvial fan deposits in Neogene basins that
surround the core of the structure (Fig. 3).

4.2 **Methods for Example I**
We collected samples from 6 sites distributed around Sierra Nevada, from all structural positions, around the
massif in unburied Plio-Pleistocene fan deposits that range from poorly cemented to unconsolidated (Sanz de
Galdeano and Vera, 1992; DR Table 1; Fig. 3).   The ages of the deposits sampled were determined from published
geologic maps (IGME-1:50,000 scale) and bridged the temporal gap between the late Miocene age metamorphic
fabrics and the present day deformation field recorded by GPS geodesy and recent seismicity. At each site, three
oriented samples were collected as independent blocks.  Before removal from the outcrop, most blocks were
hardened with a diluted (~50%) aqueous solution of sodium silicate (Fig. 4). In the laboratory, 2-3, oriented cubes
($8cm^3$) were cut from each block using non-magnetic Teflon knives and enclosed in standard cubic paleomagnetic
boxes. The anisotropy of magnetic susceptibility (AMS) was determined with an Agico Kappabridge KLY-3S at
Lehigh University. To determine magnetic mineralogy, a heating stage under the presence of an argon atmosphere
and a cold stage accessory to the Kappabridge was used.

**4.3 Results for Example I**
Results from heating and cooling experiments show a complicated magnetic mineralogy composed of 100%
ferromagnetic (magnetite or hematite) to 100% paramagnetic mineralogy (clays and iron-rich micas; Fig. 5). Since
the kinematic interpretation of each of the specimen is the same regardless of magnetic mineralogy, the details of
each specimen are not important for subsequent analysis. There is no correlation between the bulk magnetic
susceptibility ($k_m$) and the anisotropy of the magnetic ellipsoid ($P_J$), so a comparison of the principal axis of
susceptibility across the various structural positions around the Sierra Nevada massif specimens can provide useful
kinematic information (Fig. 6a).  Nearly all AMS ellipsoids are characterized by a low anisotropy degree ($P_j$) and
oblate ellipsoid shape (T) (Jelinek, 1981; Fig. 6b). The AMS axes determinations record nearly the same axis
orientations. At all sites around Sierra Nevada, $k_3$ is nearly orthogonal to bedding. The principal elongation axes
means is preferentially oriented NNE-SSW to NE-SW (Fig. 3). The orientation of the site mean magnetic
susceptibility axes, $k_1$, is horizontal or very shallowly plunging to the NE or SW (Fig 3).

**4.4 Discussion of Example I**



The AMS principal axes show a consistency between sites (Fig. 3), so we combine the susceptibility axes
orientation data in Figure 7.  These combined data suggest that during deposition the phyllosilicate grains were
oriented with their basal planes parallel or slightly imbricated to the depositional surface. Compaction during
dewatering and lithification amplified the initial oblate depositional fabric and was coincident with the formation of
the tectonic fabric. Regardless of the magnetic mineralogy of the specimens, a well-clustered minimum
susceptibility axis ($k_3$) is present, which we interpret as a compaction fabric in these sedimentary deposits. The
possibility of a primary depositional current fabric (imbrication) is unlikely because of an independent paleocurrent
study on clast imbrication at Site 3 and Site 4, which shows an eastward rather than westward transport direction
during deposition (Carrigan et al., 2018).
Irrespective of the structural position around the Sierra Nevada massif, all sites show a preferred orientation
of $k_1$. The mean principal axis of maximum susceptibility is preferentially oriented at 030°-210° (Figs.7 and 8). We
interpret this as a tectonic fabric due to the tight clustering of $k_1$ and $k_2$, the relationship between $k_1$ and strike of
dipping bedding at sites SN1, SN4, and SN6, and the lack of influence from sedimentary processes. In specimens
dominated by phyllosilicate grains it is difficult to create a strong lineation by aligning grain crystallographic axes,
however, an intersection lineation between slightly rotated clay grains orthogonal to a shortening direction has been
observed (Henry, 1997; Parés et al., 2007; Martín-Hermández and Ferré, 2007; Borradaile and Jackson, 2010). The
orientation of $k_1$ is consistent with the present day GPS velocity field, being oriented almost perfectly orthogonal to
the direction of convergence of the Betic Cordillera to stable Africa (Nubia; Fig. 2; Gutscher et al., 2012), in good
agreement with the mineral lineations recorded in the massif's core (Martinez-Martinez and Soto, 2002), and the
Neogene brittle extensional structures and recent seismicity (Mancilla et al., 2013) in the orogen (Fig. 2). Because of
the low strains and the orthogonal relationship between contractional and extensional principal directions it is not
possible to distinguish the uplift processes of the Sierra Nevada massif with our results. The AMS ellipsoid
orientations, mineralogic stretching lineation from the core of the Sierra Nevada massif, the nearby GPS velocity
field, and recent fault slip, all have orientations consistent with the same strain field (Fig. 8). The principal
elongation direction is interpreted to have persisted across different structural levels from Miocene time to the
present (> 10 m.y.).

**5 Example II: Northern Apennines, Italy**
**5.1. Geologic Setting of the Northern Apennines**
The northern Apennines are an accretionary fold and thrust belt (Bally, et al., 1986) where crustal
deformation, rock uplift, and topographic growth result from the ongoing subduction of Adria beneath Europe
(Picotti and Pazzaglia, 2008; Carminati and Doglioni, 2012).  The Apennine orogenic wedge initiated ~ 30 Ma along
the southern flank of the Alps (Le Pichon et al., 1971), and has grown at variable rates through the Neogene
dependent on the flux of mass imbricated from the subducting plate (Picotti and Pazzaglia, 2008).  Rapid rollback of
Adria with respect to Europe results in retreat and stretching of the upper plate, forming a wide zone of back arc
crustal extension.  The Apennine wedge started to become emergent ~ 4 Ma (Picotti and Pazzaglia, 2008) uplifting
and exposing paired compressional and extensional deformation fronts near the trench and in the hinterland



respectively, with the structural transition near the topographic culmination of the range (D'Agostino et al., 2001;
Carminati and Doglioni, 2012). Balanced cross-sections for the Apennines (Bally et al., 1986; Hill and Hayward,
1988) indicate ~130 to 150 km of subduction over the 30 m.y. history of the wedge, which indicates relatively slow
long-term rates at ~ 4 to 5 km/m.y. (4 – 5 mm/yr), similar to the GPS geodetic rates (Devoti et al., 2008; Caporali et
al., 2011; Bennett et al., 2012).

192         The northeastern Apennines, including the Umbria-Marche target region of this research, exposes
Mesozoic-early Cenozoic carbonates and middle-late Cenozoic mixed carbonate-siliciclastic rocks folded and
imbricated into northeast-vergent thrust sheets (Fig. 9). In Marche, these thrust sheets are located with carbonate
ridges and have inferred blind thrusts in their cores (Artoni, 2013). Further west in Umbria, the thrust sheets are
dissected by both east- and west-dipping high angle normal faults (Barchi et al., 1998; Fig. 9). Ongoing thrust
earthquakes beneath the Po Plain and Adriatic Sea (Pondrelli et al., 2006; Boccaletti et al., 2011) and normal-fault
sense earthquakes beneath the high Apennines (Lavecchia et al., 1994; Doglioni et al., 1999; Ghisetti and Vezzani,
2002; Chiaraluce et al., 2017) speak to concurrent shortening and extension in the wedge.

200         The paired deformation fronts in the northern Apennines Italy are convolved with an enigmatic, but active,
east-dipping (towards Adria), 14-15 km deep detachment called the Alto-Tiberina fault, that projects to the surface
west of the Apennine crest (Barchi et al., 1998; Pialli et al., 1998; Boncio et al., 2004; Chiaraluce et al., 2007; Eva et
al., 2014; Lavecchia et al., 2016; Fig. 9). This detachment is one of only a handful of low angle normal faults
globally that are demonstrably seismogenic (Hreinsdottir and Bennett, 2009; Valoroso et al., 2017), apparently in
contradiction to frictional fault reactivation theory that predicts that slip on low angle normal faults as extremely
unlikely (reviewed in Collettini, 2011). Most of the destructive seismicity in the high Apennines tends to nucleate
on west-dipping high angle normal faults that are antithetic to and sole into this east-dipping detachment (Galadini
and Galli, 2000; Boncio et al., 2004; Roberts and Michetti, 2004). The most destructive seismicity, including the
2016-17 earthquake sequence, is tightly focused along the highest crest of the Apennines where it is co-located with
young, underfilled, extensional basins, high angle normal faults that rupture the surface (Fig. 9), and geomorphic
evidence for an east-marching drainage divide. It is not known if the infrequent, but large historic earthquakes east
of the divide are indicative of new blind normal faults that have nucleated on the detachment, represent active
shortening, or alternatively are responding to a different stress field.

214         Imbricated foredeep and wedge-top basins contain a time-transgressive range of poorly consolidated
deposits that span the extensional and compressional regimes. Conceivably, shortening fabrics could be recorded in
lithofacies at the base of one of these basins when it was formed and filled in the shortening part of the wedge, only
to be superseded by stretching fabrics in overlying lithofacies as the basin was translated westward and into the
extending part of the wedge. Adriatic slope transverse rivers (Alvarez, 1999) traverse both the extending and
shortening parts of the wedge and contain Pleistocene alluvial deposits representing an AMS geodetic snapshot of
the current crustal strains. Published AMS data from the thrust belt shows strike-parallel (NE-SW and horizontal)
extension that is perpendicular to compression and shortening directions (Caricchi et al., 2016). To confirm these
data towards the southeast and to better locate the kinematic transition region between the contracting and extending



regions of the overlying Eurasian plate, we sampled AMS data in Oligocene and younger units, including
Quaternary deposits in a NE-SW oriented corridor across the thrust belt (Fig. 9).

**5.2 Methods for Example II**
Sampling in the Apennines was designed to identify the location of the modern extensional front. Field
collection and specimen preparation occurred like Example I from Spain, with unconsolidated samples being
hardened with sodium silicate before or just after orienting and removal from the outcrop (Fig. 4). We collected
samples from 17 sites from sedimentary rocks and poorly consolidated sediments from Late Eocene to late
Pleistocene age, with a focus on late Miocene-Pliocene argillaceous marine deposits (DR Table 1). The Italian
specimens were prepared and rock magnetic data was acquired in the Archeomagnetism Laboratory at CENIEH
(Spain).  The AMS of the collected specimens was measured on a MFK1-FA Kappabridge (AGICO Instruments), a
fully automated inductive bridge, at a frequency of 976 Hz and a field of 200 A/m.  Analysis software (Saphyr6, by
AGICO) creates a complete susceptibility tensor. Rock magnetic measurements included isothermal remanent
magnetization (IRM) acquisition experiments up to 1T and hysteresis curves to determine the relative contribution
of ferromagnetism and paramagnetism to the total susceptibility tensor. These experiments were carried out with a
Vibrating Sample Magnetometer (VSM; Micromag 3900).

**5.3 Results of Example II**
Samples from the Apennines have variable magnetic mineralogy and include a wider range of lithologies and
ages than the Betics sampling. Samples from sites AP2 and AP7 (Bisciaro Fm.) are dominated by diamagnetic
calcite and negative mean susceptibility, which precludes any meaningful analysis of the AMS axes orientations. At
most other sites, axes orientations were interpretable and the $k_1$ orientation is shown in Figure 10 for spatial
comparison. At 1T field, the magnetization was not fully-saturated, indicating the presence of hematite in addition to
lower coercivity magnetite as the dominant ferromagnetic components (Heller, 1978). Still, the bulk magnetic
susceptibility is dominated by paramagetism as revealed by the hysteresis curves (Fig. 11). The contribution of
paramagnetism suggests that the measured magnetic fabric can be used as a proxy for phyllosilicate grain's
preferred orientation, therefore, the AMS principal axes are indicators of the orientation of the strain axes orientation
(e.g., Soto et al., 2009).
Representative examples of AMS fabrics are shown in Figure 12.  The mean susceptibility shows no positive
correlation with the shape parameter or anisotropy degree (T, $P_j$; Figure 13).  Similar to the data from Spain, the
AMS ellipsoids from the Italian specimens indicate low $P_j$ values, revealing a low degree of grain shape preferred
orientation and low strains. The AMS axes distribution are particularly clear in specimens of the argillaceous and
semi-consolidated Pliocene Argille Azzurre Fm.  At all sites, $k_1$ axes orientations are shown as a function of rock
formation, as well as the sites in which $k_3$ is perpendicular to bedding (Fig. 10). All interpretable specimens from the
Apennine Range samples, including the Pleistocene fluvial deposits, generate a site mean AMS fabric consistent
with contraction and shortening in the wedge.





### 5.4 Discussion for Example II

Irrespective of sample age, we interpret AMS ellipsoids that have the magnetic lineation in a NW-SE orientation as recording contraction as this is the main trend of the fault traces and strike of bedding and topography (Fig. 10). The $k_1$ axis orientation is orthogonal to the rock transport and crustal shortening directions as recorded in GPS geodesy data and seismology (Fig. 9). A few sites do not provide interpretable kinematic results. The calcareous marls of the Bisciaro Fm. (AP2, AP7) has a poorly formed AMS fabric. In these specimens, the mean susceptibility is negative and dominated by diamagnetism, most likely calcite. The absence of a compactional fabric in carbonate dominated specimens (AP2, AP7) likely indicates that these sediments lithified by cementation soon after deposition.

In general, the distribution of the principal axes of the AMS ellipsoid does not significantly vary with stratigraphic age or structural position. For example, the oldest specimens collected from Eocene-middle Miocene marls and Pliocene siliciclastics rocks (AP6, AP14, AP17), uniformly show AMS fabrics consistent with contractional deformation of the orogenic wedge (Fig. 10). Most importantly, sites collected from thrust structures that are currently in an extending regime (AP11, AP12, AP13) implies that either the AMS fabrics was locked after the original deformation due to the high strain required to rotate grain pairs, or that subsequent extension has not affected the previous AMS fabric. (e.g., Larrasoña et al., 2004). The same is true for middle and late Miocene siliciclastic deposits astride the Marche ridge (AP3, AP9) where the current orientations of crustal stresses from fault and earthquake data are ambiguous. Pliocene and Pleistocene samples from near the toe of the orogenic wedge show an orientation consistent with ongoing shortening (AP4, AP5, AP8). Wegman and Pazzaglia (2009) also report ongoing shortening in this region as evidenced by fluvial terrace folding above the Filottrano thrust, which we cross at the location of AP4.

The kinematic transition zone in central Italy aligns with the topography, the seismicity (Pondrelli et al., 2006) and the GPS geodesy (Bennett et al., 2012; Fig. 9). Our AMS data does not improve on the location of the transition zone because of the lack of samples from Plio-Pleistocene deposits directly northeast of the drainage divide (Fig. 10). Unfortunately, the one Pleistocene river terrace deposit northeast of the divide (AP10) has indeterminate axes. As such, our AMS results are not able to support the idea that there is an apparent rotation of the principal compressive stress between the Adriatic coast and the Marche ridge associated with wedge-scale pore-pressure variations (Peacock et al., 2017). Furthermore, the AMS is unable to determine the stress field responsible for the large historic earthquakes in the region between the drainage divide and the Marche Ridge. If earthquakes in the region are related to blind normal faults with tips breaking up-section from the Alto-Tiburina detachment (Fig. 9), a possible rationale is that according to extensional critical wedge theory (Davis et al., 1983), a wedge with a taper greater than some critical value is unable to slide over its basal detachment until sufficient wedge thinning on connecting faults reduces the surface slope and wedge taper below the critical value (Xiao et al., 1991). Suitable deposits do outcrop in this critical region, so additional field work and AMS analyses may yet bear light on this problem.

### 6 Conclusions



The AMS technique provides an effective way to identify both modern and paleo-kinematics from
sediments and sedimentary rocks largely independent of the magnetic mineralogy of a specimen. Stratigraphically
controlled AMS measurements are a deep-time, paleogeodetic technique that can be combined with structural
geology, GPS geodesy, and seismic data to collectively describe the kinematics of active orogens and to better
understand the nature of seismic hazards. In both the Betic Cordillera (Example I) and northern Apennines (Example
II), weak but well-organized penetrative AMS fabrics were recovered from young unconsolidated and unburied
rocks that could not be analyzed with more traditional methods. In the Betic Cordillera we established a long-term
consistency to the strain field from the Late Miocene to the present from unburied, young deposits around Sierra
Nevada. For the northern Apennines all studied sites, regardless of site's stratigraphic age, ubiquitously record NW-
SE oriented $k_1$ axes orientations, irrespective of structural position. Contractional strains in the most southwest-
located samples are likely locked into the rocks and do not record superposed penetrative extension. At any case, the
recovered magnetic fabric orientation successfully determined the kinematics of an area near the synorogenic
surface, in the still contracting orogen toe region.
**Author Contribution**
Anastasio, Parés, and Berti conceived of the Spanish project and completed sampling, sample preparation,
measurement, and analyses.  Anastasio and Pazzaglia conceived of the Italian project. Anastasio, Pazzaglia,
Montanari, and Karnes completed the Italian sampling. Anastasio and Parés prepared the Italian specimens,
measured the samples, and analyzed the results. Anastasio, Pazzaglia, Fisher, Berti, and Kodama analyzed results
and drafted figures for the manuscript. Anastasio and Pazzaglia wrote the first draft of the manuscript and edited
each subsequent draft. Parés, Kodama, Berti, and Montanari edited multiple drafts of the manuscript.  Anastasio
completed the final edits.
**Competing Interests**
The authors all declare that they have no conflict of interest.
**Special Issue Statement**
This paper is intended for the special issue on "Tools, data and models for 3D seismotectonics: Italy a key
natural laboratory" Rita De Nardis, Massimiliano Porreca, Ramon Arrowsmith, Luca De Siena, Beatrice Magnani,
Frank Pazzaglia, and Federico Rossetti, editors.

**7 Acknowledgements**
The authors thank Andrea Rodriguez Rubio, Alondra Jimenez Perez, Isabel Hernando Alonso of CENIEH
for laboratory assistance and the Association "Le Montagne di San Francesco" for logistical support during the
sampling campaign in the Umbria-Marche Apennines. Agico is acknowledged for Anisoft software and Lisa Tauxe
is thanked for PmagPy software used to analyze the AMS data presented here. Anastasio thanks CENIEH and Parés
for hosting his academic leave during the fall 2019 semester.





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

**Figure Captions**

Figure 1. Topography and bathymetry of the western Mediterranean showing (a) the Betic orogen, southern Spain
and (b) the northern Apennine Mountains, Italy. Elevation data from GEBCO 30sec data.
https://www.gebco.net/data_and_products/gridded_bathymetry_data/

Figure 2. Geodetic, paleogeodetic, and earthquake focal mechanism data from southern Spain. Generalized geology
(from Azañon et al.,2015), focal mechanism solutions for normal faults (from Mancilla et al, 2013), mineral
lineations [short red lines] (from Martinez-Martinez and Soto, 2002), results from 10-years of observed velocity
GPS permanent [black arrows, with uncertainties] (from Gutscher et al., 2012), and campaign [yellow arrows and
uncertainties] (from Koulali et al., 2011), stations in an African (Nubia) fixed reference frame. SN = Sierra Nevada.
Bathymetry color depths as in Fig. 1. Elevation data from 30 m SRTM NASA JPL. NASA Shuttle Radar
Topography Mission Combined Image Data Set. 2014, distributed by NASA EOSDIS Land Processes DAAC,
https://doi.org/10.5067/MEaSUREs/SRTM/SRTMIMGM.003.


Figure 3. Simplified geologic map showing sample sites around the Sierra Nevada massif, southern Spain. Lower
hemisphere stereographic projection of AMS determined principal axes, $k_1$-red squares, $k_2$, green triangles, $k_3$, blue
circles. Bedding orientation shown along with axes orientation uncertainties. Elevation data from 30 m SRTM



NASA JPL. NASA Shuttle Radar Topography Mission Combined Image Data Set. 2014, distributed by NASA
EOSDIS Land Processes DAAC, https://doi.org/10.5067/MEaSUREs/SRTM/SRTMIMGM.003.
Figure 4.
Examples of specimen collection from poorly cemented samples. (a) a sampling surface is carved in a massive
sandstone of the upper Miocene Laga Fm., northern Apennines (b) the same is done on a subhorizontal layer of a
poorly cemented, fine calcareous sandstone from an upper Middle Pleistocene fluvial terrace exposed in a wine
cellar at the Geological Observatory of Coldigioco, northern Apennines. Both samples were hardened with a dilute
sodium silicate solution. Three to four oriented blocks were collected from each sampling site. Samples were
oriented with a Brunton compass and located with a handheld GPS receiver, labeled, and photographed.
Figure 5. Magnetic mineralogy of Sierra Nevada specimens. (top) Low temperature (MS vs T)
measured on a KLY-3s Kappabridge. Data in red and paramagnetic modeling in green indicating the proportion of
the magnetic susceptibility carried by paramagnetic grains. Results from all measurements indicate that the magnetic
susceptibility of the Spanish samples varies from being dominated by paramagnetic to ferromagnetic mineral grains.
The kinematic interpretation is the same in all cases. (bottom) High temperature (MS vs T) measurements showing
heating from room temperature (20°C) to 700°C  and subsequent cooling back to room temperature. All six plots
show evidence of the ferromagnetic mineral magnetite (Curie Temperature of 580°C). A lower temperature phase is
indicated in site 5, possibly maghemite. Site 7 shows the formation of additional magnetite during heating because
of the much stronger susceptibility upon cooling. Heating curves are in red and cooling curves in blue.
Figure 6. (a) Plot of mean susceptibility ($K_m$) with respect to ellipsoid shape, (T). Oblate shapes are positive T
whereas prolate shapes are negative T. The specimens are color coded by site and consistent with Fig. 3.  The lack
of correlation between ellipsoid shape and susceptibility strengthen the conclusions based on site comparisons we
present here. (b) Jelinek diagram of Sierra Nevada specimens colored by site and consistent with Fig. 3. All AMS
measurement have low anisotropy (less than 12% Pj,) and nearly all specimens are oblate (T>0). T and Pj are
calculated as follows: if n1=ln(t1), n2=ln(t2), n3=ln(t3),where t1, t2, and t3 are the eigenvalues, then T=(2n2-n1-
n3)/(n1-n3) and P'=exp(sqrt(2[(n1-nmean)2 + (n2-nmean)2 + (n3-nmean)2]) and nmean=(n1+n2+n3)/3.
Figure 7. All Sierra Nevada massif AMS data. Lower hemisphere, stereographic projection of the principal axes of
susceptibility orientations for all specimens determined from AMS measurements in stratigraphic coordinates (Fig.
3).  Arrows outside the stereonet periphery are parallel to the mean long axis ($k_1$) orientation. $k_1$= *Maximum axis,*
$k_2$= *intermediate axis, $k_3$= minimum axis.*
Figure 8. Kinematic summary of AMS Example I. Comparison of paleogeodetic methods around the Sierra Nevada
massif, Spain illustrating the validity of AMS determined principal extension direction ($k_1$).





Figure 9. (a) Location map showing the topography, major known faults, large, historic earthquakes  (from Boncio
et al., 1998) and GPS geodetic velocities (from Hreinsdottir and Bennett, 2009) in the northern Apennine research
corridor (gray shaded box).  Elevation data from  TINITALY 10 m DEM  (Tarquini et al., 2012). Alto Tiberina
Fault (ATF), Ancona (A), Apiro  (Ap), Arezzo (Ar), Ascoli Piceno (AP), Cagli (C), Camerino (Cm), Cascia (Ca),
Fabriano (F), Foligno  (Fo), Gola di Frasassi (GdiF), Gubbio (G), Jesi (J), Macerata (M), Norcia (N), Osservatorio
Geologico Coldigioco  (OGC),  Perugia (P), Spoleto (S), Visso (V). (b)  Inset  regional  map  showing the plate
boundary and location of Fig 9a.  (c) Synthetic cross section of the region in (a) projected to the X-X' line
(modified from Chiaraluce et al., 2017). Normal faults in black, thrust faults in red, top of  Permo-Triassic
evaporites in blue, top of carbonates in green.  (d) Photo of a commonly exposed bedrock fault scarp from the
Umbrian Apennines. Fault scarps are uncommon in most of Marche.

Figure 10.  Results of AMS analysis in the northern Apennines over 1:10,000 simplified geology (from regione
Marche and Umbria, regione.marche.it; http://dati.umbria.it/) and topography. Elevation data from 30 m NASA JPL.
NASA Shuttle Radar Topography Mission Combined Image Data Set. 2014, distributed by NASA EOSDIS Land
Processes DAAC, https://doi.org/10.5067/MEaSUREs/SRTM/SRTMIMGM.003. Extensional earthquake data
compiled from Rovida et al. (2020). The presence of a tectonic fabric was determined by clustering of $k_1$
declinations outside of the expected compaction fabric. Axis certainty represents the percentage of specimens of the
total used to calculate a mean $k_1$ vector. Right Legend: 1. Holocene fill; 2. 1st order Quaternary Terrace (Qt1); 3.
2nd order Quaternary Terrace (Qt2); 4 3rd order Quaternary Terrace (Qt3); 5. Argille Azzurre Fm; 6. Scaglia Rossa
Fm; 7. Maiolica Fm; 8. Bisciaro Fm; 9. Hypothesized position of the modern extensional front based on AMS
results; 10. Thrust fault; 11. Normal fault; 12. Alto-Tiberina detachment. ; 13. Drainage divide; 14. Large historic,
but pre-instrument earthquakes of unknown origin (see Fig. 9).

Figure 11. (a) Hysteresis curves for representative samples of the studied Apennine Range geologic formations (see
location in Fig. 10). Paramagnetic susceptibility clearly dominates all the specimens as revealed by the slope of the
loops. (b) Example of a specimen where the paramagnetic contribution has been removed in order to enhance the
ferromagnetic contribution (loop in black). (c) Example of a specimen where diamagnetism dominates the total
magnetic susceptibility.

Figure 12. Lower hemisphere stereographic projection of representative sites showing representative fabric patterns
in Quaternary deposits (a) and older rocks in the Apennine foreland (b), (c), and older rocks south or the extensional
front (d). The orientation of bedding is shown when not horizontal.

Figure 13. (a) Plot of mean susceptibility ($K_m$) with respect to degree of anisotropy, ($P_j$) for the Apennine specimens.
The specimens are color coded by site. (b) Jelinek diagram of Apennine specimens, colored by site. All AMS
measurement are consistent with low strains (Pj, degree of anisotropy) and nearly all specimens are oblate (T>0). T



and Pj are calculated as follows: if n1=ln(t1), n2=ln(t2), n3=ln(t3), where t1, t2, and t3 are the eigenvalues, then
T=(2n2-n1-n3)/(n1-n3) and P'=exp(sqrt(2[(n1- nmean)2 + (n2-nmean)2 + (n3-nmean)2]) and nmean=(n1+n2+n3)/3.



**Table A1.**

| Sample | Lat | Long | Elevation (m) | Formation | Age | Composition and Texture |
|---|---|---|---|---|---|---|
| **Spain** | | | | | | |
| SN1 | 37.04972 | -3.64923 | 853 | - | Quaternary | Siliciclastic silt |
| SN3 | 36.9539 | -3.05758 | 555 | - | Quaternary | Siliciclastic silt |
| SN4 | 36.95832 | -2.99537 | 600 | - | Neogene | Siliciclastic silt |
| SN5 | 37.26138 | -3.73503 | 609 | - | Neogene | Siliciclastic silt |
| SN6 | 37.00809 | -2.56091 | 501 | - | Neogene | Siliciclastic sand |
| SN7 | 37.22960 | -3.11414 | 1037 | - | Neogene | Siliciclastic sand |
| **Italy** | | | | | | |
| AP1 | 43.34778 | 13.12132 | 462 | Ghiaia Urbisaglia Fm | Early Pleistocene | Calcareous and siliciclastic silt |
| AP2 | 43.36193 | 13.09481 | 454 | Bisciaro Fm | Early Miocene | Argillaceous marl |
| AP3 | 43.35226 | 13.11542 | 502 | Laga Fm | Late Miocene | Argillaceous silty sand |
| AP4 | 43.42590 | 13.23293 | 217 | Qt4 alluvium | Late Pleistocene | Calcareous and siliciclastic silt |
| AP5 | 43.46141 | 13.30483 | 126 | Argille Azzurre Fm | Pliocene | Siliciclastic blue-gray silty clay |
| AP6 | 43.53607 | 13.59282 | 218 | Scaglia Variegata Fm | Late Eocene | Argillaceous marl |
| AP7 | 43.55456 | 13.57438 | 215 | Bisciaro Fm | Early Miocene | Argillaceous marl |
| AP8 | 43.40956 | 13.10795 | 425 | Argille Azzurre Fm | Pliocene | Siliciclastic blue-gray silty clay |
| AP9 | 43.30225 | 13.02115 | 469 | Fm Camerino (Laga Fm) | Late Miocene | Siliciclastic argillaceous sandy silt |
| AP10 | 43.40180 | 12.96773 | 223 | Qt3 alluvium | Middle Pleistocene | Calcareous and siliciclastic silt |
| AP11 | 43.41049 | 12.58075 | 553 | Marnosa Arenacea Fm | Middle Miocene | Siliciclastic argillaceous sandy silt |
| AP12 | 43.38627 | 12.56814 | 638 | Marnosa Arenacea Fm | Middle Miocene | Siliciclastic argillaceous sandy silt |
| AP13 | 43.38261 | 12.56343 | 629 | Bisciaro Fm | Early Miocene | Argillaceous marl |
| AP14 | 43.20721 | 13.00143 | 520 | Scaglia Cinerea Fm | Oligocene | Siliciclastic and calcaerous argillaceous sandy silt |
| AP15 | 43.24922 | 12.97616 | 406 | Scaglia Cinerea Fm | Oligocene | Siliciclastic and calcaerous argillaceous sandy silt |
| AP16 | 43.51872 | 12.72748 | 500 | Scaglia Cinerea Fm | Oligocene | Siliciclastic and calcaerous argillaceous sandy silt |
| AP17 | 43.56574 | 12.80247 | 421 | Laga Fm | Late Miocene | Siliciclastic argillaceous sandy silt |








Figure 1

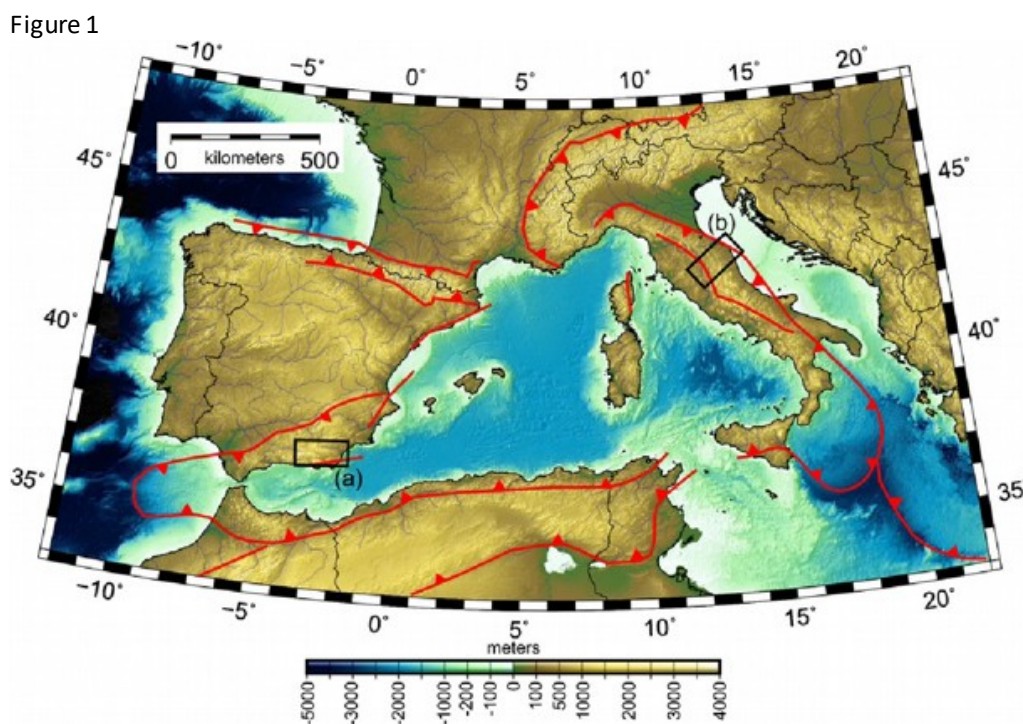




Figure 2

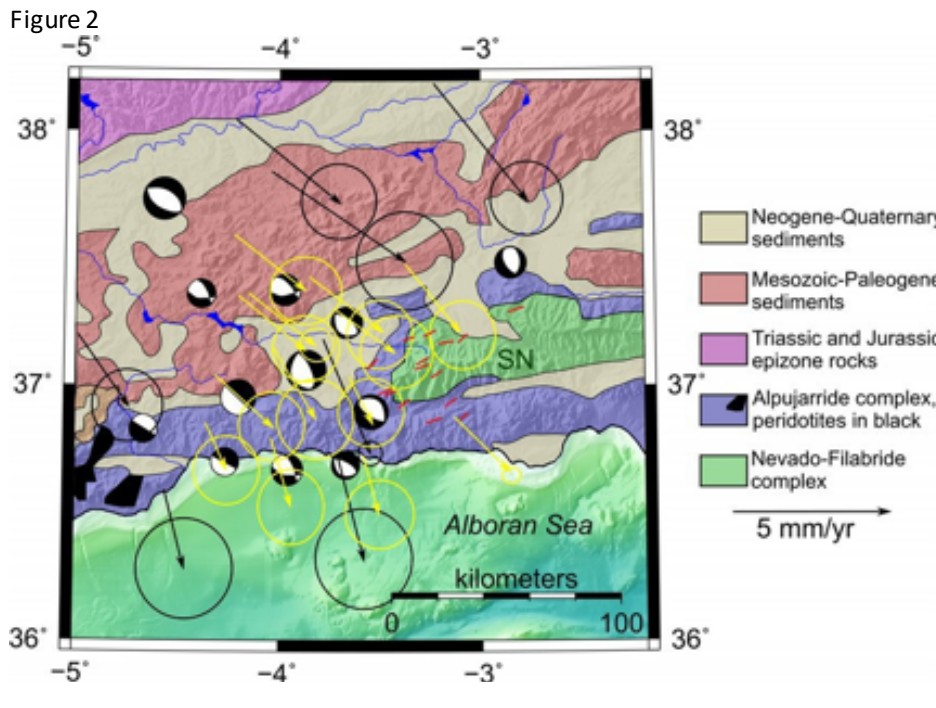




Figure 3

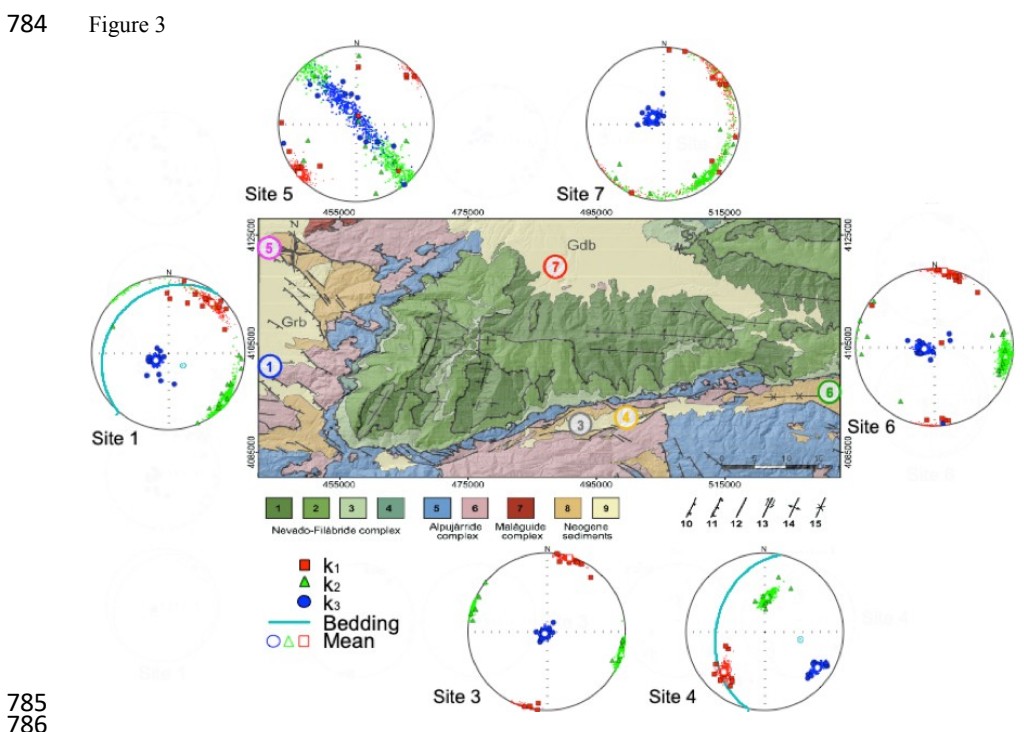





Figure 4

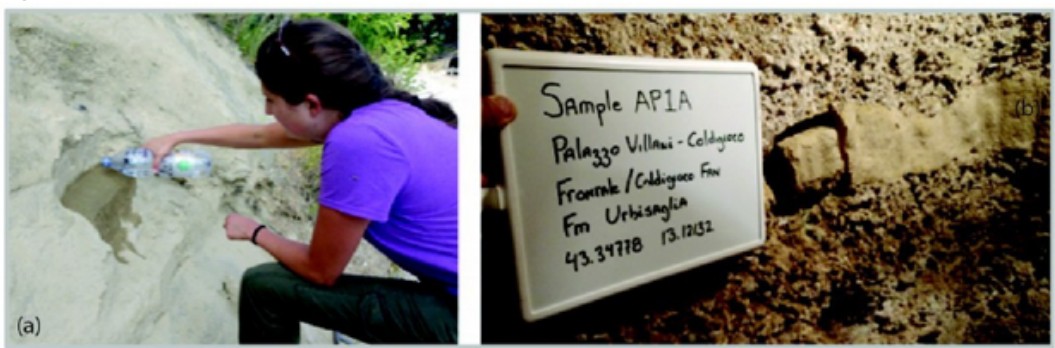






Figure 5

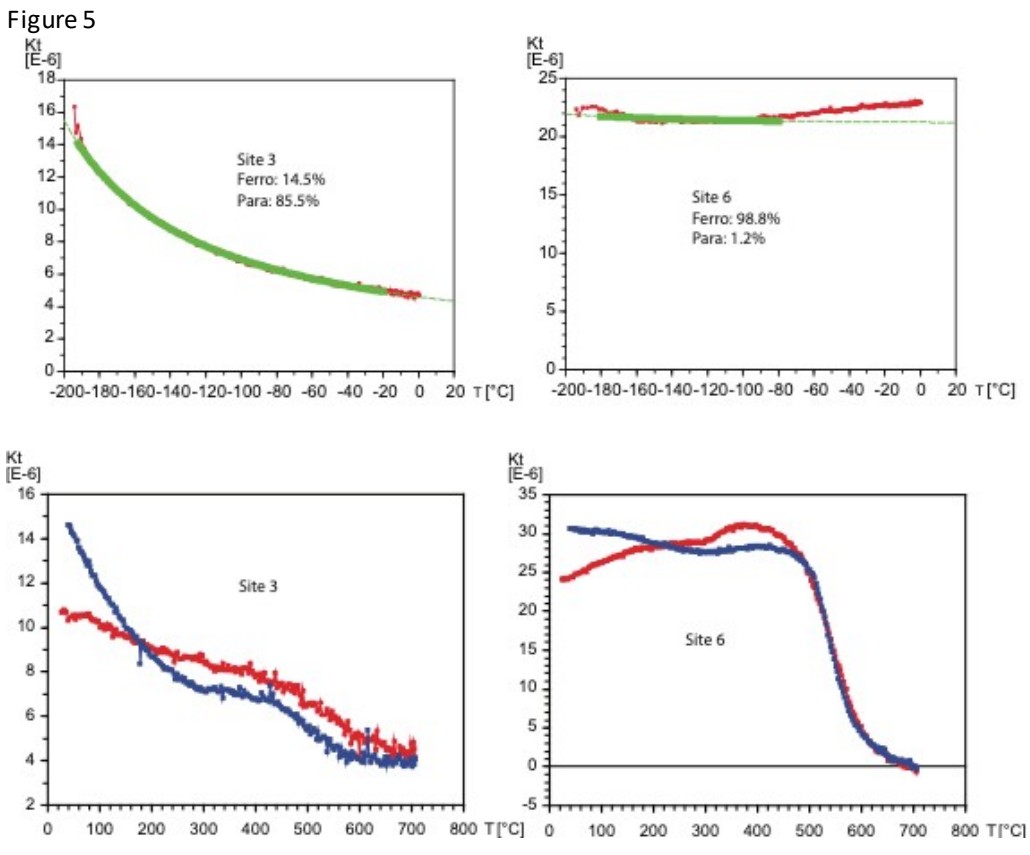




Figure 6

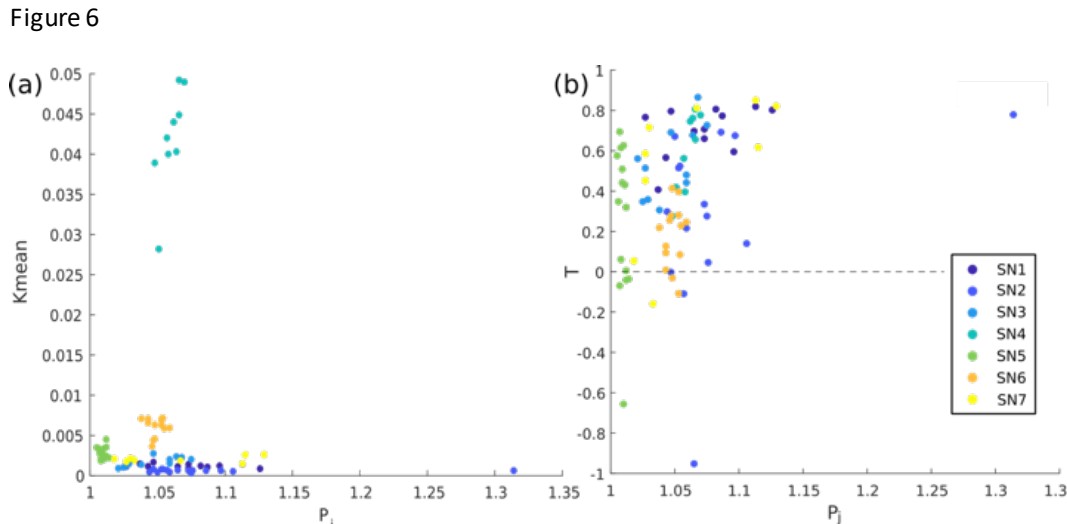






Figure 7

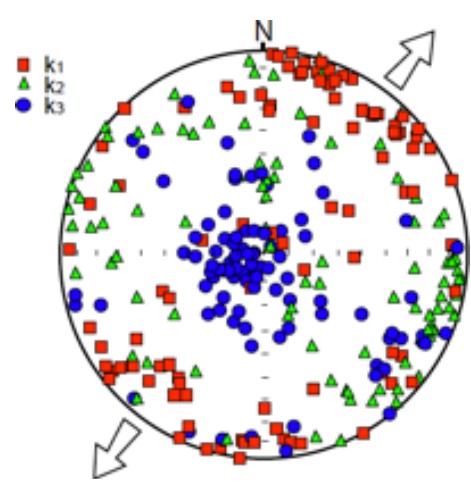




Figure 8

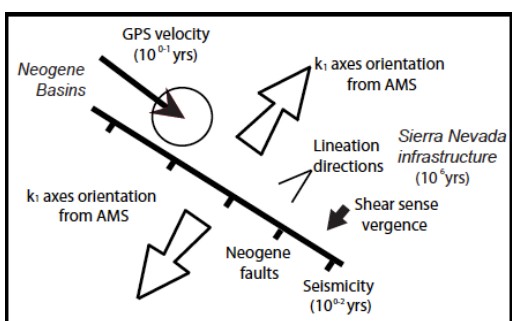




Figure 9

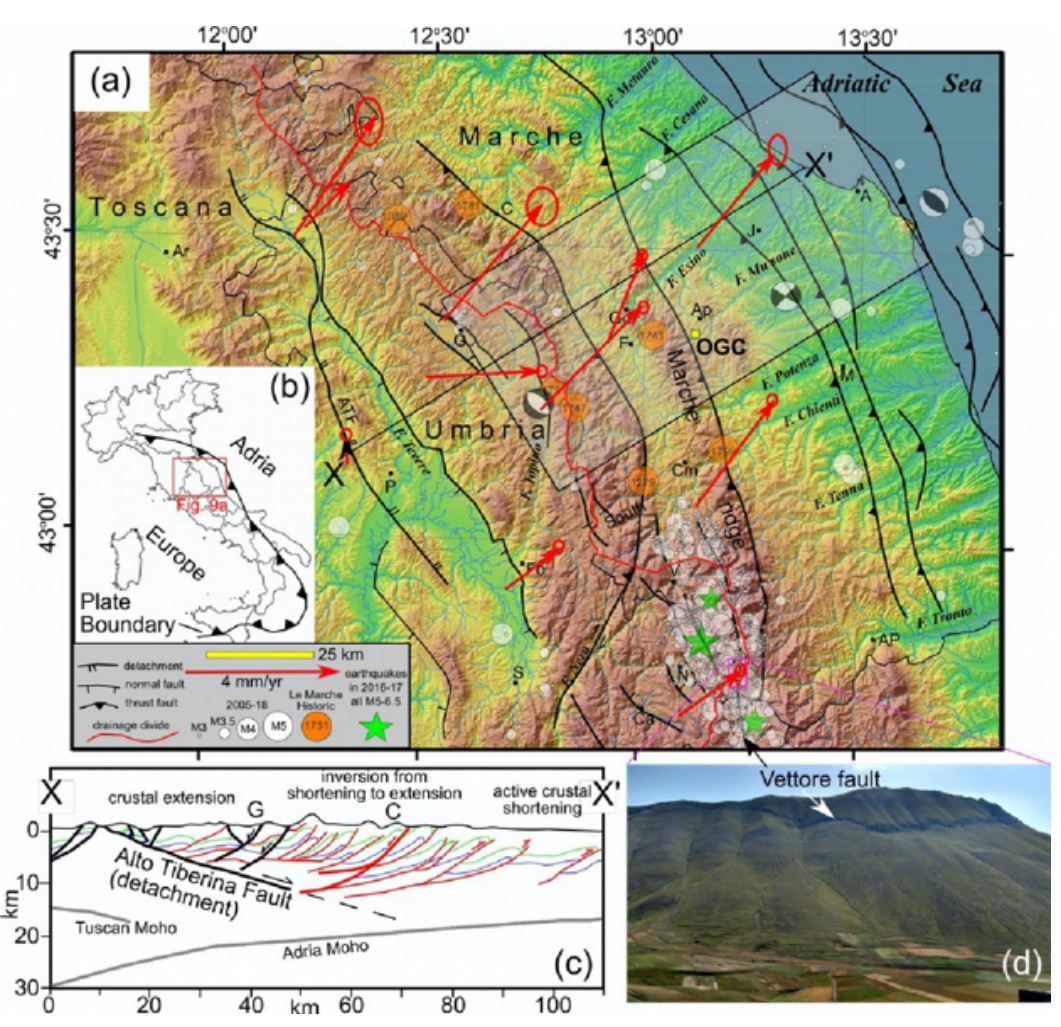



Figure 10

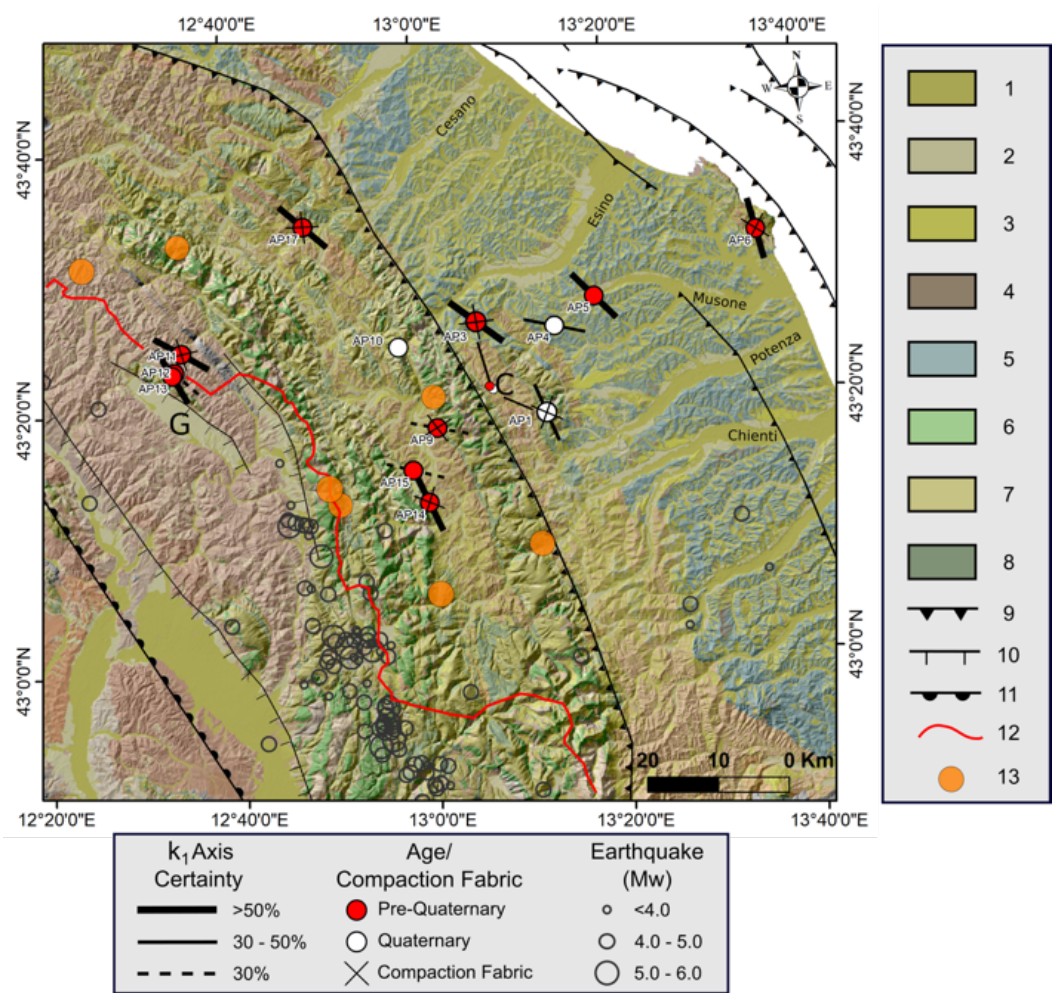





Figure 11

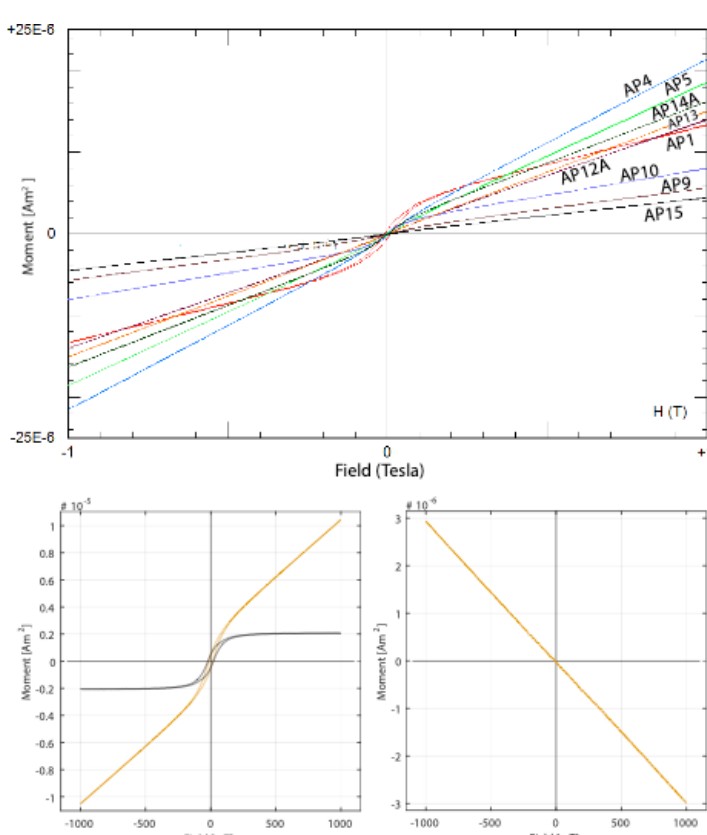




Figure 12

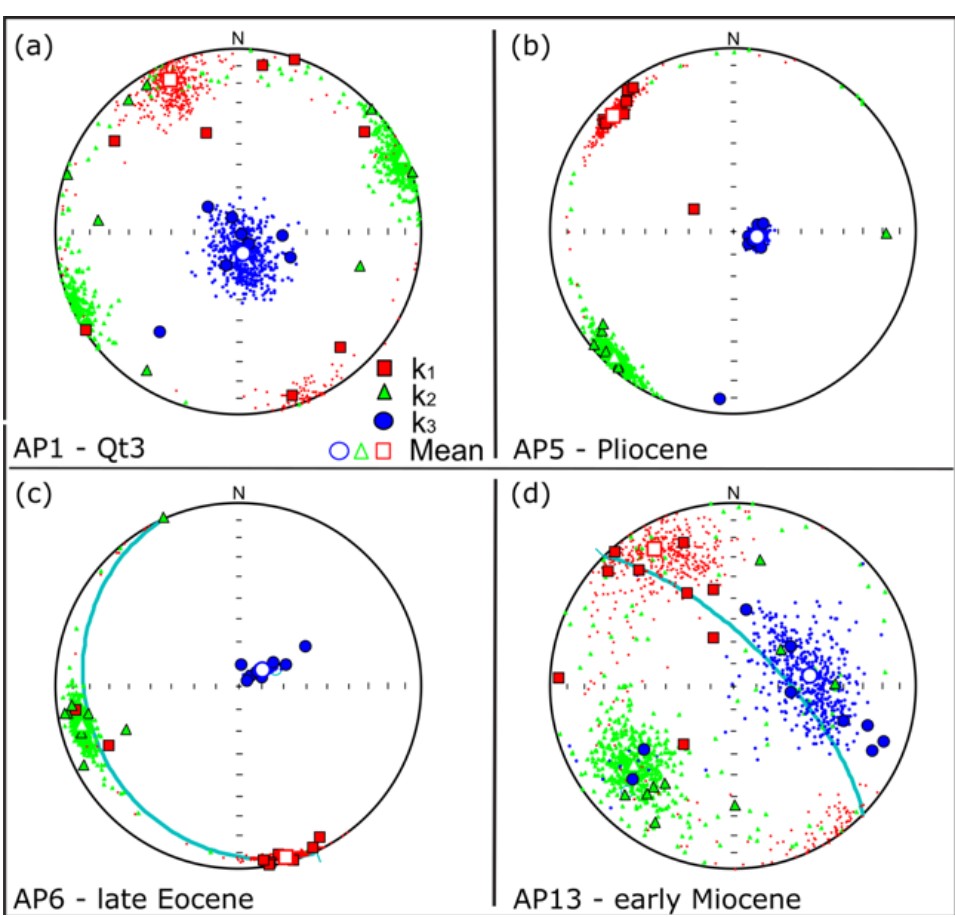




Figure 13

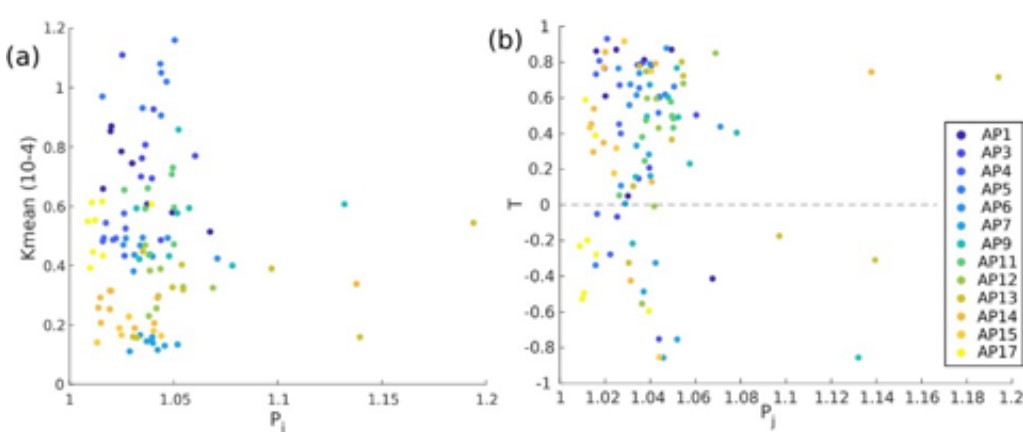
