# Peer review of "Application of anisotropy of magnetic susceptibility (AMS) fabrics to determine the kinematics of active tectonics: Examples from the Betic Cordillera, Spain and the northern Apennines, Italy."

_Solid Earth, 2020_

## Referee Comment (RC1)

[revised manuscript text omitted]

Figure 1

[Figure]

[Figure]

Figure 2

[Figure]

[Figure]

Figure 3

[Figure]

[Figure]

Figure 4

[Figure]

[Figure]

[Figure]

Figure 5

[Figure]

[Figure]

Figure 6

[Figure]

[Figure]

[Figure]

Figure 7

[Figure]

[Figure]

[Figure]

Figure 8

[Figure]

[Figure]

[Figure]

Figure 9

[Figure]

[Figure]

[Figure]

Figure 10

[Figure]

[Figure]

Figure 11

[Figure]

[Figure]

Figure 12

[Figure]

[Figure]

Figure 13

[Figure]

---

## Author Comment (AC2)

Changes to Anastasio et al., SE-2020-184-RC1
Changes based on D. Biardello's review.

Line
49      suggested change accepted, rewriting from reviewer
53      suggested change accepted, reference added
59      accepted, typo
65      deletion added, manuscript clarified
67      manuscript clarified
80      manuscript change made
93      suggested change accepted
100     suggested change accepted
134--   comment accepted
138     sentence added to caption for figure 5 outlining how paramagnetic and ferromagnetic
        components for rock magnetic mineralogy are determined.
145     There is no girdle between Kint and Kmin, this has been added to the text for
        clarification.
183     word choice change, sentence clarified
200     no change as a result of reviewer comment
228     accepted
244     no change made
247     typo fixed
248     suggested change accepted
265     suggested change accepted
267     I've clarified the manuscript text.
278     typo fixed
307     suggested change accepted
312     suggested change accepted
313     suggested change accepted
668     suggested change accepted
711     clarification sentence added to text, references cited
715     suggested change accepted
717     I've changed the figure caption to agree with the figure.
722     suggested change accepted
766     typo fixed
771     suggested change accepted
773     accepted comment, line deleted as

---

## Author Comment (AC3)

Responses to comments by Ruth Soto

General Comments
1.1 Thank you for the comment.  However, the co-authors and myself think the main scientific point of the contribution is the value of AMS measurements in young unconsolidated sediments for orogenic studies.  Therefore, we see a manuscript strengthening from multiple examples.  Previous studies that have used the Paleomagnetism laboratory at Lehigh University (i.e., Spanish data) and the Archeomagnetism Laboratory at CENIEH (i.e., Italizn samples) including:

(1) Kodama, K.P., Anastasio, D.J., Newton, M.L., Pares, J.M., Hinnov, L.A. 2010. High-resolution rock magnetic cyclostratigraphy in an Eocene flysch, Spanish Pyrenees. Geochemistry,Geophysics, Geosystems, v. 11 p. 1-22 QOAA07 doi: 10.1029/2010GC003069.

(2) Carrigan, J.H., Anastasio, D.J., Kodama, K.P., Parés, J.M. 2016. Fault-related fold kinematics recorded by terrestrial growth strata, Sant Llorenç de Morunys, Pyrenees Mountains, NE Spain. Journal of Structural Geology, v. 91, 161-176. http://dx.doi.org/10.1016/j.jsg.2016.09.003

(3) Anastasio, D.J., Teletzke, A.L., Kodama, K.P., Parés, J.M.C., Gunderson, K.L. 2020. Geologic evolution of the Peña Flexure, Southwestern Pyrenees mountain front, Spain. Journal of Structural Geology. Volume 131, Number 1, paper 103969.

Authors, Kodama, Parés, and Anastasio have an excellent track record in studies using both laboratories and we do not see the use of both laboratories as a reason not to include both field examples.

1.2 Thank you for the comment.  It is a difficult question. The magnetic lineation must be younger than the depositional age of the sediments which record it. Therefore, the timing of the lineation cannot be Miocene in age. The AMS is a low strain paleogeodetic indicator that equates to the convergence of Africa and Iberia.  It equates most uniformly with the GPS and normal fault seismicity datasets and hence is a paleokinematic indicator  The introduction ends with the sentence " In this paper, we show how AMS can extend the temporal reach of GPS geodesy back in time in orogenic studies of the Betic Cordillera, Spain and in the northern Apennines, Italy (e.g., Mattei et al., 2004; Fig. 1)".

Specific comments
2.1 Thank you for the comment.  You are correct, figure 7 was incorrect.  Figure 3 is correct and figure 7 has been corrected and replotted.  The figures now agree as to their number of specimens measured.

Conclusions, lines 297-298. In our opinion, this is a general rule that goes beyond these studies. We go on to say "Stratigraphically controlled AMS measurements are a deep-time, paleogeodetic technique that can be combined with structural geology, GPS geodesy, and seismic data to collectively describe the kinematics of active orogens and to better understand

the nature of seismic hazards. In both the Betic Cordillera (Example I) and northern Apennines (Example II), weak but well-organized penetrative AMS fabrics were recovered from young unconsolidated and unburied rocks that could not be analyzed with more traditional methods."

Technical Corrections
Comment. Balanya added to Martinez-Martinez et al., 2002 in text and references cited.
Figure 3 caption now includes geologic units.
Caption for figure 9 has been clarified.
Caption for figure 10 have been changes.  Legend now agrees with figure and caption.

---

## Editor Decision (ED1)

[revised manuscript text omitted]

(a)

(b)

Sample AP1A
Palazzo Villani - Caldigioco
Frontale / Caldigioco Fan
Fm Urbisaglia
43.34778   13.12132

[Figure]

[Figure]

[Figure]

Tilt-corrected coordinates

0

270

90

Max
Int
Min

N = 69
Equal-Area
Projection

180

[Figure]

[Figure]

[Figure]

| | |
|---|---|
| ■ | 1 |
| ■ | 2 |
| ■ | 3 |
| ■ | 4 |
| ■ | 5 |
| ■ | 6 |
| ■ | 7 |
| ■ | 8 |
| ▼▼ | 9 |
| ┬ | 10 |
| ●● | 11 |
| ～ | 12 |
| ● | 13 |

| $k_1$ Axis Certainty | Age/ Compaction Fabric | Earthquake (Mw) |
|---|---|---|
| ▬▬▬ >50% | ● Pre-Quaternary | ○ <4.0 |
| ─── 30 - 50% | ○ Quaternary | ○ 4.0 - 5.0 |
| --- 30% | ✕ Compaction Fabric | ○ 5.0 - 6.0 |

(a) AP1 - Qt3

(b) AP5 - Pliocene

(c) AP6 - late Eocene

(d) AP13 - early Miocene

■ k₁
▲ k₂
● k₃
○△□ Mean

---

## Author Response (AR2)

**Application of anisotropy of magnetic susceptibility (AMS) fabrics to determine the kinematics of active tectonics: Examples from the Betic Cordillera, Spain and the northern Apennines, Italy.**

David J. Anastasio1, Frank J. Pazzaglia1, Josep M. Parés2, Kenneth P. Kodama1, Claudio Berti3, James A. Fisher1, Alessandro Montanari4, Lorraine K. Carnes5

[revised manuscript text omitted]

| AP10   | 43.40180 | 12.96773 | 223              | Qt3 alluvium             | Middle               | Calcareous and silicicla                 |
| AP11   | 43.41049 | 12.58075 | 553              | Marnosa
Arenacea Fm   | Middle               | Siliciclastic argillaceous
sandy silt |
| AP12   | 43.38627 | 12.56814 | 638              | Marnosa
Arenacea Em   | Middle               | Siliciclastic argillaceous               |
| AP13   | 43.38261 | 12.56343 | 629              | Bisciaro Fm              | Early                | Argillaceous marl                        |
| AP14   | 43.20721 | 13.00143 | 520              | Scaglia Cinerea          | Oligocene            | Siliciclastic and calcaer                |
| AP15   | 43.24922 | 12.97616 | 406              | Scaglia Cinerea          | Oligocene            | Siliciclastic and calcaer                |
| AP16   | 43.51872 | 12.72748 | 500              | Scaglia Cinerea          | Oligocene            | Siliciclastic and calcaer                |
| AP17   | 43.56574 | 12.80247 | 421              | Laga Fm                  | Late                 | Siliciclastic argillaceous               |

---

## Editor Decision (ED2)

[revised manuscript text omitted]

(a) AP1 - Qt3

$k_1$
$k_2$
$k_3$
Mean

(b) AP5 - Pliocene

(c) AP6 - late Eocene

(d) AP13 - early Miocene